# Mu Opioid Receptor 1 (MOR-1) Expression in Colorectal Cancer and Oncological Long-Term Outcomes: A Five-Year Retrospective Longitudinal Cohort Study

**DOI:** 10.3390/cancers12010134

**Published:** 2020-01-05

**Authors:** Oscar Díaz-Cambronero, Guido Mazzinari, Francisco Giner, Amparo Belltall, Lola Ruiz-Boluda, Anabel Marqués-Marí, Luis Sánchez-Guillén, Pilar Eroles, Juan Pablo Cata, María Pilar Argente-Navarro

**Affiliations:** 1Department of Anesthesiology, Hospital Universitarii Politécnic La Fe, Avenida de Fernando Abril Martorell, 106, 46026 Valencia, Spain; 2Perioperative Medicine Research Group, Instituto de Investigación Sanitaria La Fe (IISlaFe), Avenida de Fernando Abril Martorell, 106, 46026 Valencia, Spain; 3EU-COST Action 15204, Euro-Periscope, Avenue Louise 149, 1050 Brussels, Belgium; 4Department of Pathology, Hospital Universitari i Politécnic La Fe, Avenida de Fernando Abril Martorell, 106, 46026 Valencia, Spain; 5Department of Digestive Surgery, Hospital General Universitario de Elche, Calle Almazara, 11, 03203 Elche, Spain; 6INCLIVA Biomedical Research Institute, Avenida de Menéndez y Pelayo, 4, 46010 Valencia, Spain; 7Department Medical Oncology, University of Valencia INCLIVA-Hospital Clínico de Valencia-CIBERONC, Avenida de Menéndezy Pelayo, 4, 46010 Valencia, Spain; 8Department of Anesthesiology & Perioperative Medicine, The University of Texas–MD Anderson Cancer Center, Houston, TX 77030, USA; 9Anesthesia & Surgical Oncology Research Group, Houston, TX 77030, USA

**Keywords:** colon cancer, immunohistochemistry, mu opioid receptors, perioperative opioid, perioperative medicine, disease free survival

## Abstract

Preclinical evidence has shown increased expression of mu opioid receptor 1 (MOR-1) in colorectal cancer although its association with disease-free and overall survival (DFS and OS) has not been investigated. We hypothesized that MOR-1 was overexpressed in tumor samples compared to normal tissue and this was associated with decreased DFS and OS. We carried out a retrospective study assessing the association of MOR-1 tumor expression with long-term outcomes by immunohistochemistry in normal and tumor samples from 174 colorectal cancer patients. The primary endpoint was five years of DFS. Secondary endpoints were five years of OS, the difference in MOR-1 expression between normal and tumor tissue and the occurrence of postoperative complications. Multivariable Cox regression showed no significant association between MOR-1 expression and DFS (HR 0.791, 95% CI 0.603–1.039, *p =* 0.092). MOR-1 expression was higher in tumor tissue compared to non-tumor tissue. No associations were found between MOR-1 expression and OS or postoperative complications. These findings suggest that although MOR-1 is over-expressed in colorectal cancer samples there is no association to increased risk of recurrence or mortality. Future studies are warranted to elucidate the role of cancer stage, genetic polymorphism, and quantitative assessment of MOR-1 over-expression on long-term outcomes in colorectal cancer.

## 1. Introduction

Colorectal cancer (CRC) is one of the most prevalent cancers worldwide [1]. After primary treatment of non-metastatic CRCs, 20%–40% of the patients develop recurrences, associated with poor long-term prognosis [2]. Surgical resection is the cornerstone treatment in CRC, however it leads to inflammation, activation of sympathetic nervous system, hypercoagulability, ischemia/reperfusion injury, and suppression of the immune system [3,4,5]. This stress response decreases the host capability to deal with minimal residual disease, increasing the potential risk of local recurrences or metastasis [6,7,8,9].

Type 1 mu opioid receptor (MOR-1) agonist drugs such as fentanyl, hydromorphone, and morphine are still the mainstay analgesic treatment in patients undergoing oncologic surgery [10]. Preclinical data suggest that MOR-1 is over-expressed in cancer cells and its activation is linked to cancer progression [11,12]. In addition, analgesics such as opioids may promote cancer recurrence by acting on MOR-1 [13,14,15,16,17]. 

The current evidence on the impact of MOR-1 over-expression on disease free survival (DFS) or overall survival (OS) is heterogeneous. MOR-1 over-expression is associated to poor DFS in advanced prostate cancer [18], gastric cancer [19], and hepatocellular carcinoma [5] while no association has been found in esophageal squamous cell carcinoma [20]. In CRC, MOR-1 expression has been demonstrated in vitro [21] but the association between tumor and non-tumor tissue differences in MOR-1 expression and long-term outcomes in humans has never been assessed. Furthermore, whether opioid use is associated with worse long-term outcomes in CRC patients receiving opioids is unknown [22].

The aim of this study was to investigate the association between MOR-1 expression and oncological long-term outcomes in patients with colorectal cancer. We hypothesized that MOR-1 expression is increased in colorectal cancer (versus non-tumor adjacent tissue) and is associated with shorter disease-free survival. We also evaluated the association between MOR-1 expression with overall survival and postoperative complications as well, since the latter have been associated with poor oncological outcomes [23] and can be associated with opioid usage. 

## 2. Results 

Two-hundred and twenty-eight patients were screened for eligibility, 54 patients were excluded due to stage I or postoperative stage IV classification, urgent surgery and poor sample quality, 174 patients were finally included in the study. (Appendix A). Patients’ characteristics are shown in Table 1. Samples from all 174 patients were analyzed twice blindly. Bland–Altman plot and descriptive analysis of two samples reading are reported in Appendix A. Bias was 0.66 (95% CI, 0.53–0.78) and limits of agreement were −1.04 (95% CI, −1.26−0.82) and 2.35 (95% CI, 2.13–2.57) for lower and upper, respectively. Intraclass correlation coefficient (ICC) between the two readings was very good (ICC = 0.89, 95% CI 0.86–0.91).

### 2.1. Expression of MOR-1

MOR-1 expression was higher in tumor tissue compared to non-tumor tissue from the same patient (Figure 1). Median MOR-1 expression was 3.5 (95% CI, 2.5–4.5) for tumor tissue and 2 (95% CI, 1.5–2.5) for control tissue (difference 1.50, 95% CI 1.49–1.99, *p* < 0.001). The correlation between MOR-1 expression and oncological features is shown in Appendix A. MOR-1 expression was associated with a higher number of metastatic lymph nodes and with stage III. No other significant correlations were observed. 

### 2.2. Association between MOR Expression and Long-Term Outcomes

The Kaplan–Meier analyses are reported in Figure 2. No significant differences were found for DFS or OS (log rank test *p* = 0.81 and *p* = 0.62, respectively). 

Thirty patients (17.2%) experienced a recurrence during the follow-up period and 29 (16.6%) patients died during follow-up. Univariate analysis showed a HR of 0.85 (95% CI 0.68–1.06, *p* = 0.152) for DFS and a HR of 0.88 (95% CI 0.70–1.11, *p* = 0.270) for OS. 

Similarly, complete cases multivariable Cox regression (Table 2) showed no significant association between MOR-1 expression, DFS (HR 0.791, 95% CI 0.603–1.039, *p* = 0.092) and OS (HR 1.023, 95% CI 0.784–1.335, *p* = 0.869, Figure 3). Analysis after missing values imputation yielded no significant association between MOR-1 expression and DFS and OS (Table 2). Among the covariables included in the model after the selection process by penalized regression only carcinoembryonic antigen (CEA) value at diagnosis was significantly associated with shorter DFS (HR 1.811, 95% CI 1.245–2.635, *p* = 0.002) and number of metastatic lymph nodes with OS (HR 1.482, 95% CI 1.110–1.978, *p* = 0.008). A sensitivity analysis carried out adding chemotherapy and cancer stage showed no significant changes in the effect estimate (Appendix A).

### 2.3. Association between MOR-1 Expression and Postoperative Complications

MOR-1 expression was not associated with occurrence of complications in the first 28 postoperative days both in univariate (OR 0.838, 95% CI 0.630–1.105, *p* = 0.214) and multivariable logistic regression (Appendix A).

## 3. Discussion

The main findings of this study can be summarized as follows: In patients with colorectal cancer (stage II-III), (1) expression of MOR-1 receptor was higher in tumor tissue than in normal tissue and (2) this was not associated with shorter DFS or OS.

Increased MOR-1 expression in cancer tissue have been consistently reported in the literature [5,18,19,20]. Moreover, previous in vitro results showed a higher expression of MOR-1 in colorectal cancer tissue than in normal mucosa tissue [21]. The results of our study are in line with these data. On the other hand, the association of MOR-1 over-expression with clinical outcomes is not clearly established with some trials reporting benefits [5,18,19] while others do not [20]. It is difficult to compare our results with preceding studies due to tissue specific considerations and methodological issues as previous data come from other organs′ cancers [5,18,19,20]. Moreover, our study included patients with non-advanced cancer stages, while previous studies frequently including advanced or metastatic cancer disease and MOR-1 over-expression could be a reflection of this advanced stage without any causal relationship. In addition, other factors could influence MOR-1 expression in tumor. For instance, MOR-1 increased expression has been recently linked to intraoperative opioid use [17] and this could explain differences in results. This hypothesis, however, could not be tested in our study since it requires a baseline preoperative assessment of MOR-1. 

The method of MOR-1 expression assessment is another source of heterogeneity that hinder comparisons with previous data. Some trials used IHC [18,20] while other relied on real-time quantitative polymerase chain reaction (RT-qPCR) [5,19]. These techniques target different cellular components and although clinical studies supported some correlation [24,25], results are not completely interchangeable. Moreover, different IHC scoring have been used and the technique is dependent on the pathologist interpretation. While other studies frequently used a dichotomic score and offer scant details on how they specified such dichotomy, we chose to employ a more gradual scale and carried out repeated blinded assessment of IHC staining. 

MOR-1 is encoded by the *OPRM1* gene and polymorphism in the gene locus have been described [26]. Single nucleotide polymorphism A118G have been previously linked with a reduced sensibility to exogenous opioids [27] and decreased cancer specific mortality probably due to the decreased immunosuppression associated with the G allele. Studies in breast and esophageal cancer patients found that the GG and GA alleles provided significant survival benefit compared to the AA allele [28,29,30]; it was hypothesized that a G allele increased sensitivity to endogenous opioid peptides [31,32]. In our study, we did not assess genetic polymorphism, and this could have contributed to our results, although any allele-specific in colorectal cancer patients remains to be elucidated. 

Preclinical investigations appear to indicate that the role of MOR-1 agonists is cell type-, dose- and time-dependent. Morphine has been found to be a suppressor of cells′ metastatic behavior [33], inhibitor adhesion molecules (ICAM-1) expression in endothelial cells [34]. Moreover, chronic administration of morphine inhibited tumorigenesis and metastasis [35] and reduced liver metastasis in animals [36]. Yet, other authors showed that morphine at a concentration of 100 nM stimulated the release of urokinase type plasminogen activator, a factor known to promote metastasis [21] and that the activation of MOR was associated with a significant increase in the release of interleukin-8 [37]. We found no association between total perioperative 96 h opioid use with DFS or OS. In previous studies equivalent morphine consumption has contradictory impact upon disease free survival. While it decreased survival on early stages of lung cancer [38] had no impact on colorectal or esophageal cancers [30,39,40]. 

This study has several strengths. It is the first trial that implement and strictly follow a prespecified analysis plan based on the REMARK benchmark methodology for this type of studies. Furthermore, we used an IHC score that cover all grades of staining without gaps and analyzed it as an ordinal variable without information loss due to dichotomizing process thus maximizing the power of our analysis. Moreover, the evaluation of MOR-1 expression was done by blinded repeated readings. Finally, we thoroughly collected potential confounders and analyzed the associations with rigorous controlled methodology. 

Some limitations have to be nevertheless acknowledged: (1) The retrospective design; (2) the low rate of events which limits the statistical power of any association; (3) the restricted analysis to stage II or III, non-advanced cancer; (4) the lack of evaluation the *OPRM1* gene variant polymorphism and (5) only perioperative opioid use was recorded; (6) no software for IHC evaluation. CRC adjuvant treatment is guided by an individualized recurrence risk stratification based on oncological features such as stage and genomic testing. Despite promising preliminary results in other cancer types, to date, MOR-1 IHC expression could not be implemented in adjuvant therapy guidance stratification in colorectal cancer stage II or III.

## 4. Materials and Methods 

This was an investigator-initiated retrospective single center study, conducted according to a protocol reviewed and approved by the Spanish Drugs Regulation Agency on 4 May 2018 and the Institutional Review Board of the Hospital Universitari I Politécnic la Fe, Valencia, Spain on 27 June 2018. The study was registered at clinicaltrials.gov (identifier: Clinical trials-NCT03601351) and conducted in accordance with the Declaration of Helsinki on ethical principles for medical research in human subjects, adopted by the General Assembly of the World Medical Association (1996). 

### 4.1. Study Population

Patients were eligible for participation if (a) the scheduled colorectal surgery occurred between January 2010 and December 2013; (b) they were age > 18 years and (c) and had suspected colorectal cancer for stage II/III. Exclusion criteria were: (a) Non oncologic colorectal surgery; (b) emergency or unplanned surgery; (c) and colorectal cancer for stage I or IV. Patients with poor quality histological samples were not included in analysis. Patients’ follow-up was five years from the day of surgery and all data were obtained from electronic clinical records.

### 4.2. Primary Outcome

The main outcome of this study was to evaluate the impact of MOR-1 expression by immunohistochemistry (IHC) on patients′ disease free survival (DFS) five years after surgery. 

### 4.3. Secondary Outcomes

Secondary outcomes included: (a) Differences in MOR-1 expression in tumor and non-tumor tissue; (b) association between MOR-1 expression and oncological features; (c) type of recurrence; (d) overall five-years survival; and (e) any postoperative complications until postoperative day (POD) 28. 

### 4.4. Definitions

DFS was calculated according to the National Cancer Institute definition as the length of time after primary treatment (in our study surgery) that the patient survives without any signs or symptoms of cancer progression. 

OS was defined the period of time starting from the date of the initial surgery to the time of death any cause or the last date of follow-up if no events were documented. (https://www.cancer.gov/publications/dictionaries/cancerterms?cdrid=44023)

Postoperative complications were registered and graded according to European Perioperative Clinical Outcome (EPCO) definitions [41].

### 4.5. Data Collected

MOR-1 immunohistochemistry studies were performed on paraffin-embedded human histological tissues of colorectal adenocarcinoma. In each case, we selected a sample with colorectal adenocarcinoma and a normal colonic sample. 

Human MOR-1 immunohistochemistry procedure: For antigen retrieval, sections were heated in an Envision Flex buffer (pH = 9) for 20 min and incubated for 30 min at room temperature with a mouse monoclonal MOR-1 antibody (1:100) (Acris^®^). Slides were developed for 10 min with 3,3′-diaminobenzidine chromogen and counterstained for 10 min with hematoxylin. The quantification of MOR-1 expression in human colon samples was done by microscopic evaluation of MOR-1 immunoreactivity carried out by one experienced pathologist. The observer performed two separate blinded assessments to evaluate for variability. The standard operation procedure (SOP) for IHC analysis is described in Appendix B. Immunostaining was read in a semi quantitative manner. Positive staining for MOR-1 were defined as those showing brown signals in the cell cytoplasm, nucleus, or membrane. The staining intensity was scored as “0” (no staining), “1” (weakly stained), “2” (moderately stained), or “3” (strongly stained). The percentage of cell positivity was scored as “0” (<5%, negative), “1” (5%–25%, sporadic), “2” (25%–50%, focal), or “3” (>50%, diffuse). The expression of MOR-1 was scored by adding the intensity staining scores and the percentage area positively stained, producing a total range from 0 to 6. Immunostaining control was tested successfully in the central nervous system tissue sample without MOR-1 expression. After the first immunostaining reading, the same pathologist conducted a second assessment to minimize interindividual variability. If good concordance was observed the final reading was used for analysis, otherwise a median score was calculated. 

Other variables recorded were: Gender; age; American Society of Anesthesiologists (ASA) physical status; arterial hypertension; diabetes mellitus; history of cigarette smoking; preoperative plasma total protein; anesthetic technique used (intravenous versus halogenated); epidural anesthesia use; amount of opioid drugs administered in the first 96 postoperative h (in oral morphine equivalents [42]; intraoperative remifentanil use; blood transfusion in the first 96 postoperative h; duration of surgery; neoadjuvant radiotherapy; neoadjuvant chemotherapy; adjuvant chemotherapy or radiotherapy; preoperative hemoglobin value; stage II or III cancer (%); need for reintervention; MOR-1 expression in non-tumor tissue; carcinoembryonic antigen value at diagnosis; carbohydrate antigen 19-9, number of positive lymph nodes. 

### 4.6. Sample Size Calculation

To the best of our knowledge, there is no published data in the literature on correlation between MOR-1 and DFS rate in colorectal cancer. Thus, we performed our calculation on another digestive tract cancer [19]. Based on published data on MOR-1 expression and mortality in a gastric cancer population and assuming that subjects with positive expression of MOR-1 in the neoplastic tissue had a risk ratio of 2.5 to suffer an event (with a standard deviation of 0.6) we estimated that to detect a statistically significant difference in a sample of 170 patients with a 5-year recurrence rate of 20% and an alpha error of 5% (0.05), power of 80%, and a censorship rate of 10%.

### 4.7. Analysis Plan

The analysis plan was specified before patients’ data retrieval or data analysis. Data are reported as counts and proportions or means (standard deviation, SD) or medians [25th–75th percentiles] depending on their distribution. Normality of distributions was assessed by inspection of quantile-quantile plots. Logarithm transformation was carried out if severe skewness was observed in any variable distribution. This was performed for carcinoembryonic antigen level at diagnosis and number of positive lymph nodes. 

The preliminary analysis on MOR-1 IHC differences between tumor and non-tumor tissue was carried out by paired-sample Wilcoxon rank sum test. The association between MOR-1 IHC expression and oncological features was assessed by Spearman rank correlation (ρ), or Goodman Kruskal’s gamma statistic. 

The association between MOR-1 IHC expression and DFS and OS was evaluated using the Kaplan–Meier survival curve and the Log-Rank test. For this analysis only, MOR expression was dichotomized and was defined as positive when tumor tissue had a higher expression than non-tissue tumor in a same patient’s samples and negative otherwise while the rest of analysis is carried out assessing MOR-score with the predefined ordinal scale. Moreover, a univariate estimation of association between MOR-1 IHC score and both DFS and OS was tested with the Cox model after checking for proportional risk assumption and residuals. If scaled Schoenfeld residuals plot and test did not fulfill proportional risk assumption a parametric model was fitted choosing the best fitting distribution by Akaike information criterion (AIC) [43]. 

In addition, a multivariable Cox regression model was estimated to control for potential confounding factors. Variable selection was carried out through Elastic Net with the alpha and lambda parameter estimated by cross-validation. The variables that entered the selection process are detailed in the Appendix A. Furthermore, a sensitivity analysis was performed estimating a multivariable Cox models adding chemotherapy and percentage of stage III tumor to the covariables selected.

The relationship between MOR-1 IHC score and complications at 28 postoperative days was assessed by univariate and multivariable logistic regression with variable selection process carried out with Elastic Net with same methodology as for disease free and overall survival analysis (see Appendix A for full details).

For outcome analysis (DFS and OS), cases with missing values > 5% in any covariable were included in the analysis using multiple imputation methods. The hazard ratios were derived from the pooled average effect across 10 augmented datasets, with the confidence intervals and significance tests taking into account the uncertainty of the imputations. The multiple imputation was performed by the mice package from R software (version 3.5.0, R Foundation for Statistical Computing, Vienna, Austria).

Statistical significance level will be set at *p* < 0.05. All analysis will be performed with R software.

## 5. Conclusions

MOR-1 expression is increased in colorectal cancer tissue but there is no association with lower five years DFS or OS. The results from this study did not support MOR-1 IHC expression incorporation in colorectal cancer recurrence risk stratification markers. More investigations are warranted to evaluate the role of MOR-1 over-expression, perioperative opioid use, and long-term oncological outcomes in colorectal patients.

## Figures and Tables

**Figure 1 cancers-12-00134-f001:**
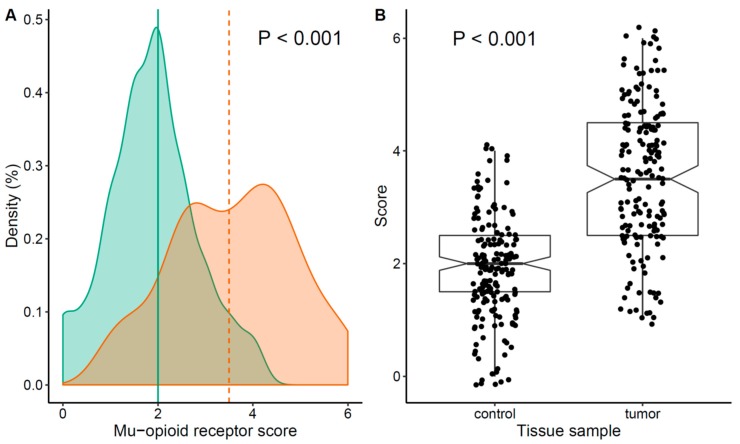
Type 1 mu opioid receptor (MOR-1) expression: (**A**) Probability density plot of MOR-1 score, Green: Normal tissue, Orange: Tumor tissue; (**B**) scatterplot and box plot of score distribution by type of sample.

**Figure 2 cancers-12-00134-f002:**
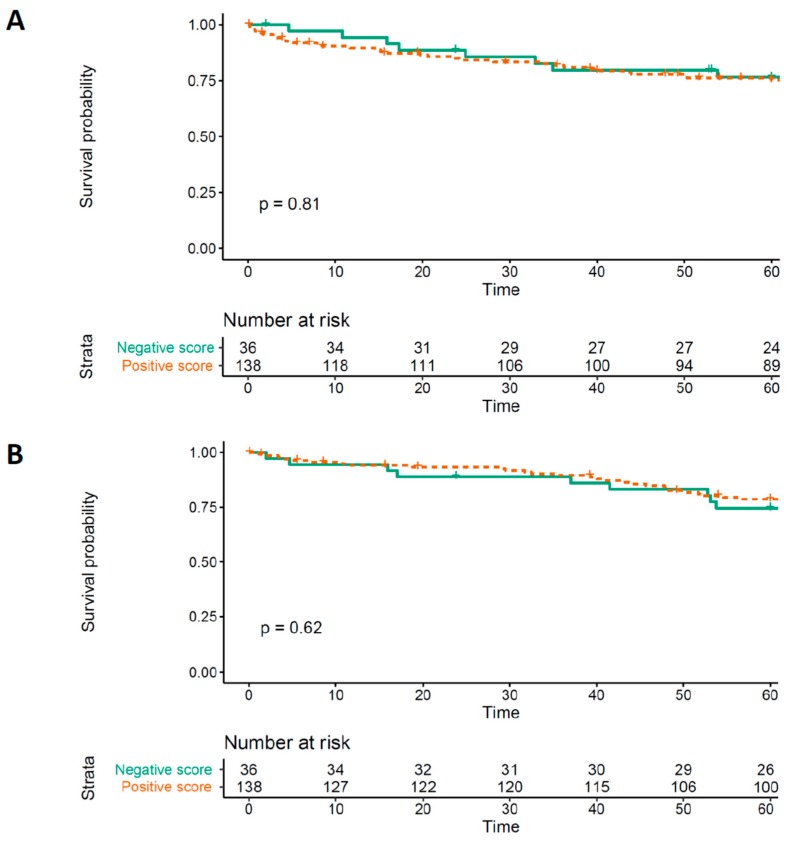
(**A**) Kaplan–Meier curve assessing MOR-1 expression effect on Disease free survival (DFS). (**B**) Kaplan–Meier curve assessing MOR-1 expression effect on overall survival (OS). MOR-1 score is dichotomized as positive when tumor tissue had higher expression than non-tissue tumor in the same patient’s samples and negative otherwise. The curves are fitted on data with imputed missing values. MOR-1: Type 1 mu opioid receptor.

**Figure 3 cancers-12-00134-f003:**
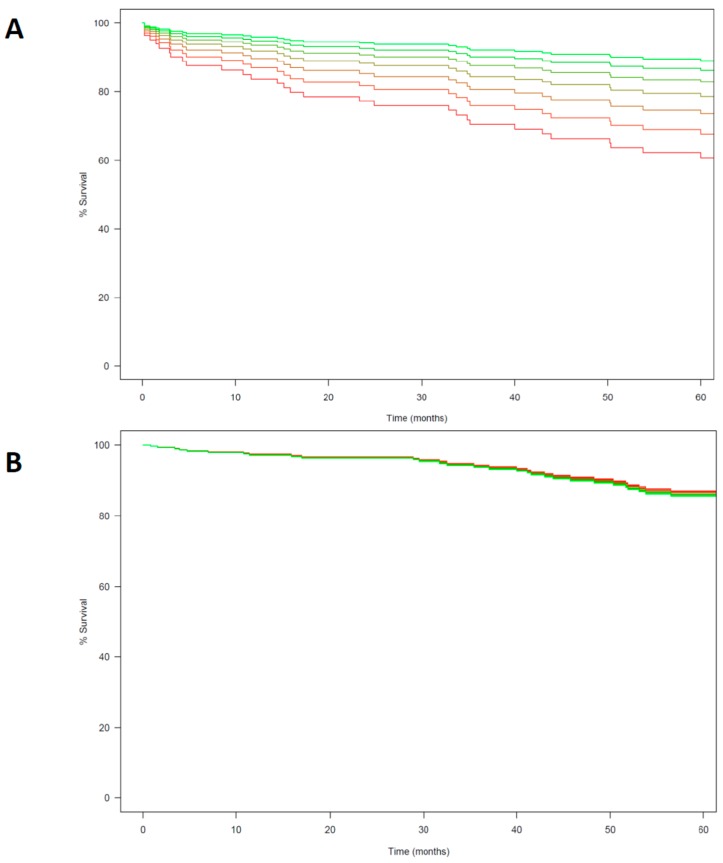
(**A**) Multivariable Cox model curve estimation for Disease free survival (DFS). (**B**) Multivariable Cox model curve estimation for Overall survival (OS). MOR-1 score is analyzed as an ordinal variable with seven levels (from 0 to 6). Different score is showed in colors from green to red with green representing a score of 0 and red a score of 6.

**Table 1 cancers-12-00134-t001:** Patients’ characteristics

Variable	*N* = (174)
Age (years)	70.5 (11.4)
Gender (female)	42.0% (73)
Complication in the first 28 postoperative days	18.4% (32)
Anesthetic agent (*N* = 166)	
Halogenated	75.3% (125)
Intravenous	20.5% (34)
Both	4.2% (7)
Intraoperative remifentanil perfusion (*N* = 169)	47.9% (81)
First postoperative 96 h total opioid dose	76.43 (34.76)
Intraoperative epidural analgesia (Yes)	16.1% (28)
Red blood cell transfusion in the first postoperative 96 h	30.5% (53)
CEA value at diagnosis (*N* = 163) (U∙mL^−1^)	2.60 [1.60–5.10]
Surgical duration (min)	217.52 (88.22)
Preoperative total proteins (g∙dL^−1^)	7.00 [6.00–7.00]
Preoperative Hemoglobin value (g∙dL^−1^)	12.03 (2.07)
Number of affected lymph nodes	0 [0–2]
Preoperative chemotherapy (Yes)	10.3% (18)
Preoperative radiotherapy (Yes)	9.8% (17)
Postoperative chemotherapy (Yes)	50.0% (87)
Postoperative radiotherapy (Yes)	1.7% (3)
ASA score (*N* = 157)	
1	7.6% (12)
2	54.8% (86)
3	33.8% (53)
4	3.8% (6)
HTA (Yes)	54.6% (95)
Diabetes Mellitus (Yes)	19.5% (34)
Reintervention Yes)	6.3% (11)
Readmission (Yes)	3.4% (6)
Dukes (*N* = 153)	
A	1.3% (2)
B	51.0% (78)
C	46.4% (71)
D	1.3% (2)
Cancer Staging (III)	44.8% (78)
Ca 19–9 value at diagnosis (U∙mL^−1^) (*N* = 124)	11.1 [5.3–18.5]
Resection margins (R+) (*N* = 135)	19% (25)
Tumoral tissue differentiation (*N* = 169)	
Poor/Undifferentiated	12.0% (20)
Moderately differentiated	78% (132)
Well differentiated	10.0% (17)

Values are reported as mean (standard deviation) or percentage (*N*) or median [25th–75th percentile] as appropriate. *HTA*: Arterial hypertension; *CEA*: Carcinoembryonic antigen; *ASA*: American Society of Anesthesiology; Ca 19-9: Carbohydrate antigen 19-9; R+: Positive resection margin.

**Table 2 cancers-12-00134-t002:** Multivariable Cox regression model for disease free survival and overall survival at five years follow-up.

Outcome of interest	Complete Cases Model			Missing Data Multiple Imputation		
Disease Free Survival Model	*N* = 135 Events = 30			*N* = 174 Events = 40		
	Hazard Ratio	Lower-Upper 95% CI	*p*-value	Hazard Ratio	Lower-Upper 95% CI	*p*-value
MOR expression	0.791	0.603–1.039	0.092	1.062	0.930–1.212	0.376
First postoperative 96 h transfusion (yes)	0.991	0.392–2.503	0.985	1.060	0.701–1.603	0.784
ASA (Reference category = 1)						
2	0.707	0.155–3.223	0.654	0.854	0.427–1.710	0.657
3	0.936	0.195–4.481	0.934	0.994	0.475–2.080	0.986
4	1.322	0.159–11.007	0.796	0.517	0.129–2.069	0.351
Preoperative Hemoglobin (g∙dL^−1^)	1.043	0.846–1.287	0.693	1.012	0.919–1.117	0.807
Number of affected lymph nodes	1.283	0.921–1.788	0.141	1.028	0.780–1.322	0.828
CEA at diagnosis (U∙mL^−1^)	1.811	1.245–2.635	0.002	1.058	0.877–1.28	0.557
Age (years)	1.010	0.970–1.052	0.638	1.005	0.987–1.022	0.591
Overall survival model	*N* = 135 Events = 29			*N* = 174 Events = 40		
MOR-1 expression	1.023	0.784–1.335	0.869	1.031	0.906–1.173	0.645
First postoperative 96 h transfusion (yes)	1.556	0.658–3.682	0.314	1.004	0.670–1.503	0.986
ASA score (Reference category = 1)						
2	0.954	0.119–7.629	0.965	0.898	0.479–1.685	0.737
3	1.948	0.247–15.357	0.527	1.072	0.538–2.138	0.843
4	2.375	0.208–27.07	0.486	0.832	0.183–3.786	0.812
Preoperative Hemoglobin (g∙dL^−1^)	0.911	0.729–1.139	0.415	1.016	0.925–1.115	0.743
Number of affected lymph nodes	1.482	1.110–1.978	0.008	0.971	0.774–1.218	0.800
CEA at diagnosis (U∙mL^−1^)	1.485	1.017–2.170	0.041	1.031	0.859–1.24	0.746
Age (years)	1.031	0.989–1.074	0.147	1.003	0.986–1.020	0.746

MOR-1 expression is introduced in both models as a 0 to 6 ordinal variable. The effect estimate is thus to be interpreted as the difference in hazard in the monitored time period when MOR-1 expression increases one level.

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
