# Peer review of "Mu Opioid Receptor 1 (MOR-1) Expression in Colorectal Cancer and Oncological Long-Term Outcomes: A Five-Year Retrospective Longitudinal Cohort Study"

_cancers, 2020, doi:10.3390/cancers12010134_

Round 1
Reviewer 1 Report
The authors investigated the prognostic impact of MOR-1 in colorectal cancer.
1.The authors explained the experimental or clinical-data based (although different cancer type) potential associations between the MOR-1 expression and survival. However, it is not quite clear that why they tried to find associations with postoperative complications. Please explain the reason in the Introduction.
Figure 2 – How they divide the patients into Negative score and Positive score groups in A) and B) ? (I think that B and C need to be changed). They explained as “MOR-1 score is dichotomized as detailed in text.” However, it is very difficult for me to find the direct explanations. If possible, please add some explanations in Figure legend directly.
Based on my understanding, the authors calculated MOR-1 using 7 levels from 0 to 6. In Table 2 (which was wrongly marked as Table 3 in 5 page of 14), how did the authors include “MOR expression” as ordinal variable or dichotomized variable as used in Figure 2? This should be clearly stated in the Foot note of Table 2.
If the authors included “MOR-1 score” as ordinal variable (from 0 to 6) in Table 2, how about using the other dichotomization such as three group comparison or different cut-offs?
The most important prognosticators in cancer patients were stage, T stage, N stage and receipt of chemotherapy. The authors should include stage or T/N stage or receipt of chemotherapy in their multivariable Cox regression model.
Figure2 in supplementary material : Please provide the descriptive results from this analysis.
How many cases did the authors include to calculate the Bland-Altman plot ?
In case of difference between the initial and second stage interpretations by pathologist, how the authors select the final IHC score per each slide?
Author Response
Reply to reviewer 1 comments manuscript ID: cancers-670670
We appreciate your remark. To our knowledge there is no experimental or clinical data on colorectal cancer, therefore the introduction is based on previous MOR–1 expression finding in other cancer types. We decided to include postoperative complications (POC) as a secondary outcome because they are also linked with poor disease-free survival and mortality and they could be related to opioid use (postoperative ileus, nausea-vomiting). We modified the introduction and provided an additional citation to better clarify this point as follows:
“The aim of this study was to investigate the association between MOR–1 expression and oncological long-term outcomes in patients with colorectal cancer. We hypothesized that MOR-1 expression is increased in colorectal cancer (versus non-tumor adjacent tissue) and is associated with shorter disease-free survival. We also evaluated the association between MOR-1 expression with overall survival and postoperative complications as well, since the latter has been associated with poor oncological outcomes [23] and can be associated with opioid usage.” (Page 2. Line 68).
Figure 2 – How they divide the patients into Negative score and Positive score groups in A) and B)? (I think that B and C need to be changed). They explained as “MOR-1 score is dichotomized as detailed in the text.” However, it is very difficult for me to find direct explanations. If possible, please add some explanations in Figure legend directly.
We apologize for the unclearness. We provided an explanation for the dichotomization procedure for the Kaplan Meier analysis in the text at Page 9, Line 265. We added it to the figure legend as well for better reading. Furthermore, we fixed the incorrect order of the panels since panel A and B are Kaplan Meier curves while panel C and D are Cox multivariable model estimations.
Now figure 2 legend read as follows:
“Figure 2. Time to event analysis. Assessment of association between MOR-1 expression in tumor sample and DFS and OS: (a) Kaplan Meier curve assessing MOR-1 expression effect on DFS. MOR-1 score is defined as positive when tumor tissue had higher expression than non–tissue tumor in the same patient's samples and negative otherwise. Missing data are imputed as detailed in the text; (b) Kaplan Meier curve assessing MOR-1 expression effect on OS. MOR-1 score is defined as positive when tumor tissue had higher expression than non–tissue tumor in the same patient's samples and negative otherwise; (c) Multivariable Cox model curve estimation for DFS. MOR-1 score is analyzed as an ordinal variable with 7 levels (from 0 to 6). The different score is showed in colors from green to red with green representing a score of 0 and red a score of 6; (d) Multivariable Cox model curve estimation for OS. MOR-1 score is analyzed as an ordinal variable with 7 levels (from 0 to 6). The different score is showed in colors from green to red with green representing a score of 0 and red a score of 6.” (Page 5, Line 104).
Based on my understanding, the authors calculated MOR-1 using 7 levels from 0 to 6. In Table 2 (which was wrongly marked as Table 3 on page 5 of 14), how did the authors include “MOR expression” as an ordinal variable or dichotomized variable as used in Figure 2? This should be clearly stated in the Footnote of Table 2.
. Thank you for this useful remark. We fixed the wrong enumeration. As for the MOR–1 expression variable, we indeed used the 0–6 levels. Thus, the estimate should be interpreted as the difference in hazard between contiguous levels. We added an explanation in Table 2 footnote as follows:
“MOR–1 expression is introduced in both models as a 0 to 6 ordinal variable. The effect estimate is thus to be interpreted as the difference in hazard in the monitored time period when MOR–1 expression increases one level.”
If the authors included “MOR-1 score” as an ordinal variable (from 0 to 6) in Table 2, how about using the other dichotomization such as three group comparison or different cut-offs?
Ideally, the MOR–1 expression effect on the outcome should be assessed as a continuum. We agree with the reviewer that there is considerable methodological heterogeneity in MOR–1 expression assessment. We tried our best to use a scale which represented all grades of expression as a useful compromise provided that we assess the dependent variable (MOR–1 expression) with immunohistochemical tools. We nevertheless feel that dichotomization in general and in this context, in particular, can lead to a loss of power and residual confounding effect (see for instance Royston, Altman, Sauerbrei Stat Med. 2006 Jan 15;25(1):127-41). Therefore, we prefer to leave the main outcome analysis as it is without further dichotomization.
The most important prognosticators in cancer patients were a stage, T stage, N stage and receipt of chemotherapy. The authors should include stage or T/N stage or receipt of chemotherapy in their multivariable Cox regression model.Thank you for the remark. To control for potential confounding effects, we performed a multivariable analysis after a thorough variable selection process with regularization methods as detailed in the supplement in Table S3 as per REMARK guidelines. We used the elastic net regularization as it helps to find a good model that takes advantage of the information in the variables and avoids the bias that you get with stepwise model selection. 24 predictors entered the selection process among which chemotherapy, stage III tumor (as a proxy for T stage as we only analyzed stage II and III colorectal cancer samples) and number of lymph nodes affected were included. The only number of lymph nodes entered the final model.
We agree however that there is no definitive selection method, thus we performed an additional sensitivity analysis including stage III and chemotherapy as covariables. The results from this sensitivity analysis did not yield any significant change in the effect estimate compared to the main analysis. We added this additional analysis full details in a new Table (Table S4) in the supplement as well as in the main manuscript as follows:
“A sensitivity analysis carried out adding chemotherapy and cancer stage showed no significant changes in the effect estimate.” (Page 4, Line 105-106).
“Furthermore, a sensitivity analysis was performed estimating multivariable Cox models adding chemotherapy and percentage of stage III tumor to the covariables selected.” (Page 9, Line 275-277).
Figure2 in supplementary material: Please provide the descriptive results from this analysis.Please find attached in the updated version of the supplement an additional Figure (Figure S3), where we report with full details the pathologist two different blinded readings. Furthermore, we report in the main manuscript the intraclass correlation coefficient (ICC) which yielded an excellent correlation between readings. We added a sentence as follows:
“Samples from all 174 patients were analyzed twice blindly. Bland–Altman plot and descriptive analysis of two samples reading are reported in Figure S2 and S3 respectively, in Supplementary Digital Content. Bias was 0.66 (95%CI, 0.53–0.78) and Limits of agreement were -1.04 (95% CI, -1.26– -0.82) and 2.35 (95%CI, 2.13–2.57) for lower and upper, respectively. The intraclass correlation coefficient (ICC) between the two readings was very good (ICC= 0.89, 95%CI 0.86 – 0.91).” (Page 2, Line 76).
How many cases did the authors include to calculate the Bland-Altman plot?
Please see also our reply to the previous commentary. Tissue samples were analyzed twice, thus in the Bland–Altman plot, we plotted the two different readings (A and B) from all 174 patients. We explicitly report the procedure in the updated manuscript in the Result section.
In case of difference between the initial and second stage interpretations by the pathologist, how the authors select the final IHC score per each slide?
Please see also our reply to commentaries 6 and 7. We agree with the reviewer that this is unclear. The agreement between the two different readings was very good, thus we use the second reading for the analysis as joining both readings with median value yielded the same scores. To better describe how we proceeded, we added a sentence in the methods section as follows:
“After the first immunostaining reading, the same pathologist conducted a second assessment to minimize interindividual variability. If good concordance was observed the final reading was used for analysis, otherwise a median score was calculated.” (Page 9, Line 238).

Reviewer 2 Report
In this article, Diaz-Cambronero et al. have investigated the link between MOR-1 expression and oncological outcomes in colo-rectal cancer. Their results have shown that MOR-1 is overexpressed in tumor tissues compared to normal tissues in a same patient but this overexpression is not associated with an impact on DFS, OS or complications in the first 28 POD even if MOR-1 expression is associated with a higher number of metastatic lymph nodes and with stage III.
To do so, the authors have first scored MOR-1 expression in tumor and non-tumor tissues by IHC from stage II/III patients and then collected patients’ characteristics and survival over 5 years to perform statistical analysis.
This retrospective study is well designed with exhaustive data even if we would have liked to know the part of opioids consumption before and late after the surgery. Regarding to the survival curves, I was wondering if the authors could find significant differences in survival by grouping patients into 2 groups : low and high MOR-1 expression. Moreover, I wonder if it would not have been relevant to analyze data by using ratio between MOR-1 expression in tumor and non-tumor tissues for a same patient. Limitations are well-detailed in the discussion. By the way, the authors suggested at the end of the discussion that MOR-1 expression level could be incorporated in therapy guidance, but they haven’t developed this part. Can they explain how the MOR-1 could be used in therapy guidance ? what kind of clinical studies should be done ? Anyway, this article is very interesting for the field of oncology and perioperative care and I recommend to accept it after minor revision. Indeed I have noticed 2 text editing :
The number of the table between line 109 and line 110 should be table 2 (and not table 3). In Figure 3 (Immunohistochemistry sample to describe scoring), a scale bar should be included into all the pictures.Author Response
Reply to reviewer 2 comments manuscript ID: cancers-670670
Thank you for your comment, we also agree that could have been an interesting point. The total amount of opioid drugs administered in the first 96 postoperative hours was indeed recorded, although it was eliminated as covariable during the selection process for the final multivariable model. Unfortunately, we could not separate these data into pre- and post-surgery administration. Furthermore, chronic opioid usage before surgery data was not available. We acknowledge this limitation in the Discussion section.
Regarding the survival curves, I was wondering if the authors could find significant differences in survival by grouping patients into 2 groups: low and high MOR-1 expression.
Please see also our reply to commentary 2 and 4 from reviewer 1. We fixed the incorrect labels for figure 2 panels. We found no difference in Kaplan Meier analysis after dichotomizing MOR–1 expression. We chose our cut–off on a biological basis considering the score as positive when MOR–1 expression was higher in the tumor tissue than in same patients' control sample. Although 'low' and 'high' expression has been used in previous studies we feel that dichotomizing, albeit conceptually useful, can lead to loss of power and unreliable results especially when no prespecified low or high expression is defined.
Moreover, I wonder if it would not have been relevant to analyze data by using the ratio between MOR-1 expression in tumor and non-tumor tissues for the same patient.
This is a very interesting point, as the real impact of MOR-1 expression could be related to different rather than the absolute tumor MOR-1 expression. This was the underlying rationale we had in mind when implementing the Kaplan Meier analysis dichotomizing between positive (those patients with greater MOR–expression in tumor tissue e.g. the positive ratio of MOR–1 expression) and negative score. We agree however that further studies are warranted on how exactly measure expression and which parameter is more useful.
Limitations are well-detailed in the discussion. By the way, the authors suggested at the end of the discussion that MOR-1 expression level could be incorporated in therapy guidance, but they haven’t developed this part. Can they explain how the MOR-1 could be used in therapy guidance? what kind of clinical studies should be done?We select stage II and III because since the best treatment approach is less clear and new markers for risk stratification are continuously been described. We assume that if MOR-1 expression can be associated with poorer oncologic outcomes, this could be incorporated as a marker for guided individualized recurrence risk stratification. We modified the paragraph in the discussion to better explain this concept. The paragraph now reads:
“CRC adjuvant treatment is guided by an individualized recurrence risk stratification based on oncological features or markers such as CEA or lymph nodes invasion. Despite promising preliminary results in other cancer types, to date, MOR-1 IHC expression could not be implemented in therapy guidance stratification in colorectal cancer stage II or III.”. (Page 7, Line 179)
Indeed, I have noticed 2 text editing: The number of the table between line 109 and line 110 should be table 2 (and not table 3).Please see also our reply to commentary 2 from reviewer 1. We fixed Tables numeration in the updated version of the manuscript.
In Figure 3 (Immunohistochemistry sample to describe scoring), a scale bar should be included in all the pictures.Thanks for this suggestion. We implemented a 100µmscale bar.Figure 3 legend is also modified accordingly: “Immunohistochemistry sample to describe scoring. All pictures are at 10X magnification. Scale bar = 100µm. (a) Central nervous system tissue control;(b) Score 0;(c) Score 1;(d) Score 2;(e) Score 3;(f) Score 4;(g) Score 5;(h) Score 6;”

Round 2
Reviewer 1 Report
There are several issues need to be solved.
1-1.In legend of Figure2, the authors stated that “MOR-1 score is defined as positive when tumor tissue had a higher expression than non–tissue tumor in a same patient's samples and negative otherwise.” However, in the same paragraph, they stated that “MOR-1 score is analyzed as an ordinal variable with 7 levels (from 0 to 6).”. It looks like that they defined “MOR-1 score” using two conflicting definitions. It is very confusing. Please modify correctly.
1-2.The authors used the term MOR-1 score and MOR expression mixed in the overall manuscript. For example, in page 9 line 266, the authors stated that “MOR expression was dichotomized and was defined as positive when tumor tissue had a higher expression than non-tissue tumor ---“. I believe that the MOR expression in this sentence is same as MOR-1 score used first in legend of Figure 2. These situations might be also confusing. Please modify accordingly.
1-3. In Table 2, the authors used MOR expression grading (from 0-6) as an essential parameter. In contrast, in Figure 2, the authors compared the two dichotomized group based on the positive and negative expression. I think that the MOR expression grading (from 0-6) and the positive and negative dichotomization are totally different parameter. However, as a reader, we commonly believed that parameters used in Kaplan-Meier graph and Cox multivariable analysis are same. Thus, it could cause severe confusion in interpreting the results. I believe that this might be the main reason that have some confusion throughout the paper. Thus, if the authors wanted to keep Figure 2, I highly recommend to define the meaning of outcomes as follow in Material and method section and used appropriately (please modify based on your policy).
“We defined MOR-1 expression and MOR-1 score using two different categories. The MOR-1 expression is defined as positive when tumor tissue had a higher expression than non-tissue tumor in a same patients’ sample and negative otherwise. In addition, MOR-1 score is defined as an ordinal variables with 7 levels, which was defined in Table 3. “
2.In Figure 2-A and Figure 2-B, initial starting number of “Risk at number” section is 36 in Negative score and 138 in positive score. This meant that a total of 174 patients were included in generating Kaplan-Meier curve of DFS (Fig 2-A) or OS (Fig 2-B). As the authors stated that, there were two different models, one is “complete cases model” and the other is “Missing data multiple imputation” model. It should be stated clearly that which model was used in generating these Kaplan-Meier curves in Figure 2.
3.In Figure 2, compared with Figure 2-C, Figure 2-D had only two lines in the plot. I understand that the authors included MOR-1 score (from 0-6) in Cox model, thus the Figure 2-C had 7 lines in the plot. I think that D also had 7 plots rather than 2 plots. Please explain this.
4.The authors stated that they selected meaningful variables using elastic net penalization algorithms. In my experience, the selected variables might not be same in DFS model and OS model because the events might be different between recurrence (DFS) and death (OS). However, in Table 2, the authors used the same variables both in DFS model and OS model. Please explain this.
5.The numbers included in Table 1 have some mistakes. For example, the sum of anesthetic agent is 166, not 174. The rate of intraoperative remifentanil perfusion should be 46.5% (81/174) not 49.1%. The rate of intraoperative epidural analgesia should be 16.1% (28/174), not 16.9%. Also, in Table 1 listed in page 3, there are at least 6 wrong numbers, such as the sum of Dukes stage is only 153, not 174. It is not clear theses are derived from either the missing values or incorrect calculations. Please check again and correctly modify.
In addition, in page 4, “thirty patients (22%)” might be derived from 30/135. However, at this stage, the readers believed that these events were calculated from the whole number (174), thus they believed that the rate of 17.2% (30/174) would be correct.
6.The authors included a below paragraph in revision based on the other reviewer’s recommendation.
“CRC adjuvant treatment is guided by an individualized recurrence risk stratification based on oncological features or markers such as CEA or lymph nodes invasion. Despite promising preliminary results in other cancer types, to date, MOR-1 IHC expression could not be implemented in therapy guidance stratification in colorectal cancer stage II or III.”
Within above paragraph, as far as I know, CEA is not used as an important maker for selecting adjuvant treatment. In the current guidelines, stage and genomic testing such as microsatellite instability, KRAS or BRAF mutational status might be used to guide therapeutic treatment depending on the stages. Please check again and if you believed that CEA could be used as a guidance, you need to show a reference.
Author Response
Reply to reviewer 1 comments manuscript ID: cancers-670670-2.
In legend of Figure 2, the authors stated that “MOR-1 scoreis defined as positive when tumor tissue had a higher expression than non–tissue tumor in a same patient's samples and negative otherwise.”However, in the same paragraph, they stated that “MOR-1 score is analyzed as an ordinal variable with 7 levels (from 0 to 6)”. It looks like that they defined “MOR-1 score” using two conflicting definitions. It is very confusing. Please modify correctly.
We acknowledge that Figure 2 is possibly hard to understand since it includes a lot of data. Following this suggestion and to avoid any potential confusion that this figure may cause, we decided to split the panel into two separate figures. In the updated version of the manuscript, Kaplan Meier analysis and Cox multivariable regression are now on two separate figures (Figure 2 and 3 respectively) since they represent distinct analysis with a different treatment of MOR–1 score. The variable definition is detailed in each legend. Figure 2 and 3 now reads:
“Figure 2. (a) Kaplan Meier curve assessing MOR-1 expression effect on DFS. (b) Kaplan Meier curve assessing MOR-1 expression effect on OS. MOR-1 score is dichotomized as positive when tumor tissue had higher expression than non–tissue tumor in the same patient's samples and negative otherwise. The curves are fitted on data with imputed missing values”. (Page 5, Line 108)
“Figure 3. (c) Multivariable Cox model curve estimation for DFS. (d) Multivariable Cox model curve estimation for OS. MOR-1 score is analyzed as an ordinal variable with 7 levels (from 0 to 6). Different score is showed in colors from green to red with green representing a score of 0 and red a score of 6”. (Page 6, Line 113).
Of note, as Kaplan Meier analysis only allows for binary variable, we dichotomized MOR–1 score while in all other analysis MOR–1 score is defined as an ordinal variable with 7 levels. We detail explicitly this issue in the methods section as follows:
"The association between MOR-1 IHC expression and DFS and OS was evaluated using the Kaplan-Meier survival curve and the Log-Rank test. For this analysis only, MOR expression was dichotomized and was defined as positive when tumor tissue had a higher expression than non–tissue tumor in a same patient's samples and negative otherwise while the rest of analysis is carried out assessing MOR–score with the predefined ordinal scale" (Page 10, Line 264)
1-2. The authors used the term MOR-1 score and MOR expression mixed in the overall manuscript. For example, in page 9 line 266, the authors stated that “MOR expression was dichotomized and was defined as positive when tumor tissue had a higher expression than non-tissue tumor ---“. I believe that the MOR expression in this sentence is same as MOR-1 scoreused first in legend of Figure 2. These situations might be also confusing. Please modify accordingly.
Thank you for this useful appreciation. Please see also our reply to commentary 1–1. We assessed MOR-1 expression using a score. This score is an ordinal variable with 7 levels for all the analysis except for the Kaplan Meier analysis where we used a binary variable to define MOR–1 expression. We detailed this in the text as suggested by reviewer and hopefully made this clearer. MOR-1 score could be dichotomized or analyzed as an ordinal variable. Otherwise we review this expression all over the text and modify accordingly.
1-3. In Table 2, the authors used MOR expression grading (from 0-6) as an essential parameter. In contrast, in Figure 2, the authors compared the two dichotomized group based on the positive and negative expression. I think that the MOR expression grading (from 0-6) and the positive and negative dichotomization are totally different parameter. However, as a reader, we commonly believed that parameters used in Kaplan-Meier graph and Cox multivariable analysis are same. Thus, it could cause severe confusion in interpreting the results. I believe that this might be the main reason that have some confusion throughout the paper. Thus, if the authors wanted to keep Figure 2, I highly recommend defining the meaning of outcomes as follow in Material and method section and used appropriately (please modify based on your policy).
“We defined MOR-1 expression and MOR-1 score using two different categories. The MOR-1 expression is defined as positive when tumor tissue had a higher expression than non-tissue tumor in a same patients’ sample and negative otherwise. In addition, MOR-1 score is defined as an ordinal variable with 7 levels, which was defined in Table 3. “
We agree with the reviewer. Please see also our reply to commentary 1–1. Since joining Kaplan Meier and Cox analysis in the figure lead to the misleading impression that they were the same analysis we now clearly separate them in distinct figures. We would like also to point out that Kaplan Meier and Cox are quite different methods as the former run on a single binary predictor, it is a non–parametric method and do not yield any effect estimate but only a P-value. We added a slightly modified version of the sentence suggested by the reviewer in the methods section.
2.In Figure 2-A and Figure 2-B, initial starting number of “Risk at number” section is 36 in Negative score and 138 in positive score. This meant that a total of 174 patients were included in generating Kaplan-Meier curve of DFS (Fig 2-A) or OS (Fig 2-B). As the authors stated that, there were two different models, one is “complete cases model” and the other is “Missing data multiple imputation” model. It should be stated clearly that which model was used in generating these Kaplan-Meier curves in Figure 2.
We agree. We specifically state which data set we used (the one with multiple imputation) in the new figure legend. The caption now reads as follows:
"Figure 2. (a) Kaplan Meier curve assessing MOR-1 expression effect on DFS. (b) Kaplan Meier curve assessing MOR-1 expression effect on OS. MOR-1 score is dichotomized as positive when tumor tissue had a higher expression than non–tissue tumor in a same patient's samples and negative otherwise. The curves are fitted on data with imputed missing values." (Page 5, Line 108)
3.In Figure 2, compared with Figure 2-C, Figure 2-D had only two lines in the plot. I understand that the authors included MOR-1 score (from 0-6) in Cox model, thus the Figure 2-C had 7 lines in the plot. I think that D also had 7 plots rather than 2 plots. Please explain this.
We apologize for the improvable quality of the figure. Please see also our reply to commentary 1–1 and 2. We have now splinted the misleading panel into two separate figures which allows to bigger figures. The last panel (Figure 2–D, now figure 3–B) has indeed 7 curves which represent the effect estimate for each score level. Unfortunately, the differences between levels are really tiny for OS after taking into account the effect of confounder (which is one of the many advantages of Cox regression over Kaplan Meier method) giving the impression that only two curves are plotted. We hope that in the next figures it could be better appreciated.
4.The authors stated that they selected meaningful variables using elastic net penalization algorithms. In my experience, the selected variables might not be same in DFS model and OS model because the events might be different between recurrence (DFS) and death (OS). However, in Table 2, the authors used the same variables both in DFS model and OS model. Please explain this.
Thank you for this remark. We selected elastic net because we believed that the method had some interesting advantages for our study. Our data have almost the same number of confounders and cases and in this context regularization algorithm seem to have better performance compared to conventional stepwise methods. We agree nonetheless that there is no perfect method to select covariables for a model. In this regard, we performed an additional sensitivity analysis (Table S4) in reply to a previous commentary on round 1, adding some important covariables to the model (chemotherapy and tumor stage).
5.The numbers included in Table 1 have some mistakes. For example, the sum of anesthetic agent is 166, not 174. The rate of intraoperative remifentanil perfusion should be 46.5% (81/174) not 49.1%. The rate of intraoperative epidural analgesia should be 16.1% (28/174), not 16.9%. Also, in Table 1 listed in page 3, there are at least 6 wrong numbers, such as the sum of Dukes stage is only 153, not 174. It is not clear theses are derived from either the missing values or incorrect calculations. Please check again and correctly modify.
We apologize for the unfortunate mistakes. We fixed these errors in the updated version of Table 1.
In addition, in page 4, “thirty patients (22%)” might be derived from 30/135. However, at this stage, the readers believed that these events were calculated from the whole number (174), thus they believed that the rate of 17.2% (30/174) would be correct.
Thank you very much for this remark. We report the correct percentage in the updated version of the manuscript. The sentence now reads as follows:
"Thirty patients (17.2%) experienced a recurrence during the follow–up period and 29 (16.6%) patients died during follow–up. Univariable analysis showed a HR of 0.85 (95%CI 0.68–1.06, P=0.152) for DFS and a HR of 0.88 (95% CI 0.70–1.11, P=0.270) for OS." (Page 4, Line 95)
6.The authors included a below paragraph in revision based on the other reviewer’s recommendation.“CRC adjuvant treatment is guided by an individualized recurrence risk stratification based on oncological features or markers such as CEA or lymph nodes invasion. Despite promising preliminary results in other cancer types, to date, MOR-1 IHC expression could not be implemented in therapy guidance stratification in colorectal cancer stage II or III.”Within above paragraph, as far as I know, CEA is not used as an important maker for selecting adjuvant treatment. In the current guidelines, stage and genomic testing such as microsatellite instability, KRAS or BRAF mutational status might be used to guide therapeutic treatment depending on the stages. Please check again and if you believed that CEA could be used as a guidance, you need to show a reference.
Thank you for this useful remark, we modify text according to your suggestion. The sentence now reads as follows:
"CRC adjuvant treatment is guided by an individualized recurrence risk stratification based on oncological features as stage and genomic testing. Despite promising preliminary results in other cancer types, to date, MOR-1 IHC expression could not be implemented in adjuvant therapy guidance stratification in colorectal cancer stage II or III."(Page 7 line 178).
